# *Aedes* Mosquito Distribution along a Transect from Rural to Urban Settings in Yaoundé, Cameroon

**DOI:** 10.3390/insects12090819

**Published:** 2021-09-12

**Authors:** Borel Djiappi-Tchamen, Mariette Stella Nana-Ndjangwo, Timoléon Tchuinkam, Idene Makoudjou, Elysée Nchoutpouen, Edmond Kopya, Abdou Talipouo, Roland Bamou, Marie Paul Audrey Mayi, Parfait Awono-Ambene, Charles Wondji, Christophe Antonio-Nkondjio

**Affiliations:** 1Vector Borne Diseases Laboratory of the Research Unit Biology and Applied Ecology (VBID-RUBAE), Department of Animal Biology, Faculty of Science, University of Dschang, Dschang P.O. Box 067, Cameroon; borel_tchamen@yahoo.com (B.D.-T.); timotchuinkam@yahoo.fr (T.T.); bamou2011@gmail.com (R.B.); mayimariepaulaudrey@yahoo.com (M.P.A.M.); 2Institut de Recherche de Yaoundé (IRY), Organisation de Coordination pour la Lutte Contre les Endémies en Afrique Centrale (OCEAC), Yaoundé P.O. Box 288, Cameroon; stellanana123@gmail.com (M.S.N.-N.); idenemakoudjou@gmail.com (I.M.); edmondoev@yahoo.fr (E.K.); atalipouo@gmail.com (A.T.); hpaawono@yahoo.fr (P.A.-A.); 3Department of Animal Physiology and Biology, Faculty of Science, University of Yaoundé I, Yaoundé P.O. Box 337, Cameroon; 4Centre for Research in Infectious Disease (CRID), Yaoundé P.O. Box 13591, Cameroon; enchoutpouen2002@yahoo.fr (E.N.); charles.wondji@lstmed.ac.uk (C.W.); 5Vector Biology Liverpool School of Tropical Medicine Pembroke Place, Liverpool L3 5QA, UK

**Keywords:** *Aedes albopictus*, *Aedes aegypti*, rural, peri-urban, urban, breeding site, Yaoundé, Cameroon

## Abstract

**Simple Summary:**

Vector surveillance is key for the prevention of arbovirus disease outbreaks. In the present study, the distribution of the *Aedes* vector population between the city centre and a close rural setting was assessed. Larval mosquito collections were undertaken from November 2019 to November 2020 along a transect from the rural area to the city centre. All containers with water were inspected. Some entomological indices evaluating transmission risk were estimated. A total of 6332 mosquito larvae were collected. Different *Aedes* species were recorded, including *Ae. albopictus, Ae. aegytpi, Ae. simpsoni* and *Aedes* spp. The density of mosquitoes collected during the rainy season (4706) was high compared to the dry season (1626). *Ae. albopictus* was the most abundant *Aedes* species in the urban (96.89%) and peri-urban (95.09%) sites while *Ae. aegypti* was the most abundant species in rural settings (68.56%). *Ae. Albopictus* preferred breeding habitats were discarded tires (42.51%), whereas *Ae. aegypti* was more prevalent in plastic containers used for storing water (65.87%). High stegomyia indexes traducing a high arbovirus transmission risk were recorded. The study suggests a high frequency of *Aedes* species in Yaoundé and its neighbourhood and stresses the need for urgent action to control *Aedes* populations in the city of Yaoundé.

**Abstract:**

Introduction: The surveillance of mosquito vectors is important for the control of arboviruses diseases worldwide. Detailed information on the bionomics and distribution of their main vectors, *Aedes aegypti* and *Aedes albopictus,* is essential for assessing disease transmission risk and for better planning of control interventions. Methods: Entomological surveys were carried out from November 2019 to November 2020 in six localities of Yaoundé city following a transect from urban to rural settings: two urban (Obili, Mvan), two peri-urban (Simbock, Ahala) and two rural areas (Lendom, Elig-essomballa)—during rainy and dry seasons. All water containers were inspected. *Aedes* mosquito abundance, species distribution and seasonal distribution patterns were compared using generalized linear models. Stegomyia indexes were estimated to determine the risk of arbovirus transmission. Results: A total of 6332 mosquitoes larvae were collected (2342 in urban areas, 1694 in peri-urban areas and 2296 in rural sites). *Aedes* species recorded included *Ae. albopictus*, *Ae. aegytpi*, *Ae. simpsoni and Aedes* spp. High mosquito abundance was registered in the rainy season (4706) compared to the dry season (1626) (*p* < 0.0001). *Ae. albopictus* was the most abundant *Aedes* species in urban (96.89%) and peri-urban (95.09%) sites whereas *Ae. aegypti* was more prevalent in rural sites (68.56%) (*p* < 0.0001). Both species were found together in 71 larval habitats. *Ae. albopictus* was mostly found in discarded tires (42.51%), whereas *Ae. aegypti* was more prevalent in plastic containers used for storing water (65.87%). The majority of *Aedes* mosquitoes’ breeding places were situated close to human dwellings (0–10 m). Conclusion: Uncontrolled urbanization seems to greatly favour the presence of *Aedes* mosquito species around human dwellings in Yaoundé. Controlling *Aedes* mosquito distribution is becoming urgent to reduce the risk of arbovirus outbreaks in the city of Yaoundé.

## 1. Introduction

Arboviruses pose a serious threat to public health worldwide. Dengue, known as the most widespread arboviral disease, is responsible for more than 90 million cases and approximately 40,000 deaths yearly [1]. Other *Aedes* borne diseases, including Zika, Chikungunya, yellow fever, Rift Valley fever and West Nile, are also important public health threats [2]. During the last decades, there has been a dramatic resurgence of arboviral diseases around the World [3,4,5,6]. As the World responds to repeated outbreaks, surveillance activities and tailored control interventions are desperately needed to protect human populations [7]. 

The frequency of arbovirus cases could result from the rapid adaptation capacity of their vectors to new environments and viruses to new hosts [8,9,10]. Furthermore, the current modification of the environment by anthropogenic activities, including the exploitation of lowland area and swamps for farming and/or house construction, the storage of water in containers due to poor water supply in cities, the disposal of domestic wastes close to human habitations and emerging factors related to globalization and climate changes, are all shaping vector populations’ expansion ranges and arboviral disease transmission [11,12,13]. They are then used by *Aedes* adults females for laying eggs. Such factors are conducive to epidemics in tropical and subtropical areas, especially in sub-Saharan Africa [14]. 

Enzootic cycles of arboviral diseases are maintained by vectors such as *Ae. africanus*, *Ae. luteocephalus, Ae. simpsoni* and *Ae. opok,* whereas *Ae. aegypti* and *Ae. albopictus* are involved in both urban and rural transmission cycles [15,16]. *Aedes albopictus* was reported for the first time in Cameroon in 2001 [14,17], while *Ae. aegypti* has been reported in the country since the 1950s [18]. Nowadays, *Ae. albopictus* is well established across the country except in the north and far north regions, and tends to replace the native species, *Ae. aegypti,* which now predominates in suburban and rural areas [19,20,21,22,23,24].

Many serological investigations revealed that the number of Dengue, Chikungunya and Zika cases has significantly increased over the last several decades in Cameroon, highlighting the urgent need for a risk assessment of arboviral diseases across the country. Laboratory experiments conducted with *Ae*. *Albopictus* populations from Cameroon indicated that the species was competent to transmit Dengue, Yellow fever and Zika virus, as is the native *Ae*. *aegypti* [25,26,27]. In the absence of treatment or vaccines for most of these arboviruses, control efforts rely mainly on vectors control. Continuous surveillance of vector populations is the most reliable method, not only for monitoring vector populations dynamics [28], but also for predicting the transmission risk of arbovirus diseases to human populations [29]. 

Urbanization could potentially modify *Aedes* mosquito ecology by changing the composition and dynamics of species and increasing the abundance of their breeding sites through anthropogenic changes [30,31]. Accordingly, understanding the population dynamics and range expansion of vectors is of utmost importance for disease surveillance and control. This study explored the population dynamics of *Aedes* mosquitoes species and entomological larval indices along a transect from rural to urban areas. It also provides updated information on the distribution of *Aedes* mosquito species in Yaoundé and its neighbourhood.

## 2. Materials and Methods

### 2.1. Description of Study Sites

The study was conducted along a transect from urban to rural settings of Yaoundé (Figure 1) and included: two sites in the urban centre, Obili (3°51′26″ N; 11°29′33″ E) and Mvan (3°37′0″ N; 12°18′0″ E); two peri-urban sites, Simbock (3°51′24.768″ N; 11°32′52.872″ E) and Ahala (3°80′00″ N; 11°48′33″ E); and two rural sites, Lendom (3°57′0″ N; 11°30′0″ E) and Elig-essomballa (11°45′37.90″ N; 3°87′11.75″ E). Yaoundé is located within the Congo–Guinean phytogeographic zone, which is characterized by a typical equatorial climate with four seasons: two rainy seasons (March to June and September to November) and two dry seasons (December to February and July to August) [32]. The city has a population estimated at about 3 million inhabitants and is situated 800 m above sea level [33]. The landscape of Yaoundé is characterized by an alternation of high and lowland areas frequently used for agricultural practices.

In the rural sites, which are Lendom and Elig-essomballa (distance estimated to be 1220.38 m), houses were mostly of the traditional style, constructed with mud or wood. These villages are surrounded by a preserved primary rainforest, which provides strong vegetation with dense canopy cover, trees with holes and bamboos. Simbock and Ahala (peri-urban sites separated by 4190.37 m), situated at the city’s periphery, are characterized by residential buildings with space between houses, large roads and lowland. Obili and Mvan (urban sites, separated by 4304.73 m) are densely populated sites characterized by the exploitation of lowland area for house construction with substandard housing. The distance between urban and peri-urban sites was 4734.17 m, and from urban to rural sites it was estimated to be 4216.14 m.

### 2.2. Study Design

Entomological field surveys were conducted from November 2019 to November 2020. During field surveys, oral consent to inspect potential breeding sites was obtained from household or garage owners. *Aedes* immature stages were collected in natural and artificial breeding sites such as tree holes, dead leaves, flower pots, used tires, tanks and abandoned plastic containers. During inspections, each potential breeding place was geo-referenced with a global positioning system (GPS) and, if positive, the following parameters were registered: breeding site type, presence of larval stages, larval instars (L1–L2 or L3–L4), presence of pupae, distance of the breeding sites to the nearest house. Any container with water around houses was considered a breeding habitat. The distance between breeding habitats and houses was estimated. Aquatic habitats were classified according to the following distance ranges from the nearest house: 0–10 m, 10–50 m, and >50 m. Once collected, immature stages from each breeding site were stored in individual plastic containers (0.5 L) and were brought to the insectary of OCEAC (organization of coordination and fight against the great Endemics in Central Africa) for rearing under controlled conditions (70–80% humidity, 28 ± 1 °C). After emergence, mosquitoes were provided with 10% sucrose solution and adults were identified per breeding site under a binocular magnifying glass using morphological identification keys [34,35]. Identified mosquitoes were preserved either in silica gel or in RNA (SIGMA Aldrich, Saint Louis, MO, USA) for further molecular analysis.

Estimation of entomological indices, species richness, and mosquito abundance:

During field surveys, all water containers around houses were inspected in order to detect *Aedes* larvae and pupae. Different entomological indices were determined, including: House Index (HI)—the percentage of houses found infested with larvae and/or pupae; Container Index (CI)—the percentage of water holding containers with active immature stages; Breteau Index (BI)—the number of positive containers per 100 houses inspected. When HI > 35%, BI > 50, and CI > 20%, the area was considered at high risk of yellow fever transmission, whereas areas with HI < 4%, BI < 5 and CI < 3% were considered to be low risk for yellow fever transmission [36]. Similarly, for Dengue transmission risk in a particular area, with a value of HI < 0.1%, the area is considered low risk; a value of HI ranging from 0.1%–5% indicated medium risk and HI > 5% indicated high risk for Dengue transmission [32]. These indices have been commonly used for risk assessment and served as early warning of Dengue epidemics [34,35]. Species richness (number of species) and mosquito abundance (total number of mosquitoes collected independently of the species) were determined according to habitat type, breeding site type and seasons.

### 2.3. Data Analysis

Statistical analysis was performed using the environment for statistical computing and graphics R version 4.0.4 (“Lost Library Book” Copyright (C) 2021 The R Foundation for Statistical Computing Platform). Species richness (number of species) and mosquito abundance (total number of mosquitoes collected independently of the species) were determined according to habitat type, breeding site type and seasons. Individual-based rarefaction curves for all habitat types across seasons were constructed using the “vegan” package. Generalized Linear Models (GLM) were run (based on a chi-squared distribution using the type III sums of squares method with the package “car”) to assess the effect of habitat types (sites), breeding habitats, and seasons on mosquito species’ occurrence and distribution. Dunnett’s T3 test was used to compare the Container index between sites. The level of significance for statistical analysis was 0.05.

## 3. Results

### 3.1. Mosquito Distribution across Ecological Settings

A total of 6332 mosquitoes belonging to five genera (*Aedes, Culex, Anopheles, Toxorhynchite* and *Eretmapodites*) were collected (Table 1). Out of this number, 2342 (36.98%) were collected at urban sites, 1694 (26.75%) at peri-urban sites and 2296 (36.26%) at rural sites. A total of 5672 *Aedes* mosquitoes were collected with 4260 (75.10%) *Ae. albopictus*, 1314 (23.16%) *Ae. aegypti* and 85 (1.49%) *Ae. simpsoni* (Table 1). A few of the collected *Aedes* species (*n* = 13) that could not be identified morphologically were grouped under *Aedes* spp. *Ae. albopictus* (2214) was present at all sites and represented the most frequent *Aedes* species in urban (96.89%) and peri-urban (1512; 95.09%) sites, whereas *Ae. aegypti* was the most common *Aedes* species in rural areas (1165; 68.56%). A significant association between *Aedes* mosquito abundance, species and habitat types was recorded (*p* < 0.0001). A significant association was recorded using GLM when assessing the relationship between mosquito species’ distribution vs. sampling sites or season or breeding habitat types (Table 2).

### 3.2. Distribution of Aedes Species According to Seasons and Ecological Settings

The abundance and distribution of *Aedes* mosquito species in each ecological setting was significantly different between the dry and rainy seasons (*p* < 0.0001). A high density of *Aedes* mosquitoes species was observed during the rainy season (*n* = 4706; 74.32%) compared to the dry season (*n* = 1626; 25.67%), especially in peri-urban (93.83%) and urban areas (96.81%). When assessing species richness, a high diversity of *Aedes* species was recorded in rural areas (four species in the dry season versus three species in the rainy season) compared to the other sites where two species were found (Figure 2).

### 3.3. Types of Aedes Breeding Habitats

*Aedes* larvae were collected in a great variety of breeding sites including water containers, tires, discarded containers or plants (Figure 3). Tires (41.68%) were the most common, followed by plastic containers (33.99%). Out of the 403 breeding sites inspected; 86 (21.33%) were in urban sites, 136 (33.74%) in peri-urban sites and 181 (44.91%) in rural sites.

*Ae. albopictus* larvae were mostly found in discarded tires (42.51%) in urban and peri-urban areas while in rural settings, they were observed in plastic containers used for storing water (33.30%) (Table 2). *Ae. aegypti* larvae were also frequently found in plastic containers used to store water (65.87%) at rural sites. Significant associations between the presence of plastic barrels (*p* < 0.0001), tires (*p* < 0.0001) and “others” (sprayer, sink, wheelbarrow) (*p* = 0.0008) and *Ae. albopictus* larvae presence were recorded (*p* < < 0.0001) (Table 3).

### 3.4. Co-Occurence of Aedes Species in Breeding Sites

*Ae. albopictus* was recorded more frequently than *Ae. aegypti* and was found in 154 (70%) larval habitats, whereas *Ae. aegypti* larvae were recorded in 66 (30%) larval habitats. Both species were found together in 71 (32.27%) breeding habitats. Out of the 154 breeding habitats with *Ae. albopictus* larvae, 56 habitats had only *Ae. albopictus* and these were frequent in urban and peri-urban areas. In rural areas, *Ae. albopictus* was only recorded alone in seven habitats. *Ae. aegypti*, on the other hand, was recorded alone in 27 habitats. At several breeding sites, *Ae. albopictus* was found with other species. Indeed, different associations were observed with habitats containing two, three, four or five different mosquito species. The co-occurrence of different genera is also more frequent in rural areas compared to urban settings. Species belonging to different genera, including *Anopheles*, *Culex* and *Eretmapodites,* were detected in sympatry (co-occurrence) with *Aedes* larvae (Table 4).

### 3.5. Distance of the Breeding Sites to the Nearest House

Most *Aedes* mosquitoes’ breeding places were found close to human habitations. Over 50% of breeding sites were situated less than 10m from houses at all sites. Less than 20% of the breeding sites in urban, peri-urban and rural areas were situated above 50 m. In rural areas, up to 70% of breeding places were situated less than 10m from houses (Figure 4).

### 3.6. Estimation of Entomological Indices (Stegomyia Indices)

An estimation of *Stegomyia* indices was performed to assess the risk of arbovirus transmission in different ecological settings. High Breteau (49.38%) and Containers (62.66%) indexes were registered in urban sites (Table 5). In peri-urban settings, the container index (46.29%) was the highest index, while the house index (44.91%) was the highest in rural settings.

## 4. Discussion

This study indicated a permanent presence of *Aedes* mosquito species in Yaoundé and its neighbourhood. *Aedes* species observed included *Ae. aegypti* and *Ae. albopictus,* which are largely distributed across the country [37]. *Ae. simpsoni* was also registered in the collections conducted in rural areas. Previous studies reported the presence of this species in rural settings of Cameroon [31,38,39,40]. Some *Aedes* species were recorded but could not be identified to the species level and were termed *Aedes* spp. *Aedes* species were observed in association with culicine species such as *Culex quinquefasciatus, Cx. duttoni, Cx. antennatus, Lutzia tigripes, Cx. Culiciomayia group, Eretmapodites* sp. *Aedes* species were recorded during both the rainy and dry seasons, yet a slight increase in densities was observed during the rainy season in all three sites, supporting a dependence on seasonal conditions. *Aedes* species were observed breeding in different types of habitats including plastic or metallic containers, discarded tires, cans, tree holes and in leaves. Although *Ae. aegypti* and *Ae. albopictus* were found to prefer human made habitats, different preferences were observed for the two species. *Ae. aegypti* was found to be abundant in water storage containers (plastics and metal), especially water jars, whereas *Ae. albopictus* was highly prevalent in discarded tires, empty cans and containers. These findings were in accordance with previous studies in Cameroon [41,42] and elsewhere in central Africa [43,44,45]. Used vehicle tires have been reported as the main larval habitats and presumably one of the most productive for *Ae. Albopictus* [41,42]. *Aedes* albopictus was also reported to produce eggs resistant to desiccation; these specific adaptation characteristics could have promoted the distribution of the species across the world [46]. As a result of an overlapping geographical distribution and shared microhabitats between *Ae. aegypti* and *Ae. albopictus,* it has been proposed that competition during larval development is shaping the distribution of both species. Recent studies suggested that the invasion of most parts of the world by *Ae. albopictus* has induced a decline in the abundance of *Ae. aegypti* and could even lead to its disappearance when both of them share the same larval breeding place. A competitive displacement of *Ae. aegypti* by *Ae. albopictus* has been documented in previous studies [47]. In the Americas, in both Brazil and the USA, it was reported that competition during larval development contributed to the displacement of *Ae. aegypti* by *Ae. albopictus* in various places [48,49,50]. It is possible that interspecific interactions between adult mosquitoes, such as interspecific mating or satyrization on female *Ae. aegypti* as they mate freely with male *Ae. albopictus* in addition to males of their own species [51,52], could be shaping the distribution of the two species. Studies from the University of Florida indicated that female *Ae. aegypti* have evolved resistance to cross-mating [52,53]. It is so far not known whether interspecific mating is shaping the distribution of *Ae. aegypti* and *Ae. albopictus* in Cameroon and this deserves further investigation and regular surveillance of vector population dynamics.

In Yaoundé, *Ae. aegypti* was the predominant species in the urban environment before the introduction of *Ae. albopictus*. Since *Ae. albopictus’* introduction in the late 1990s [17], this species has now become the predominant species in both urban and peri-urban areas; whereas *Ae. aegypti* is now mostly found in rural settings. Our findings are in accordance with previous studies in the city of Yaoundé and in Cameroon [37,41,54,55]. However, although in Brazil, *Ae. albopictus* was first detected in 1986 and has now invaded almost all Brazilian states [56,57], the distribution pattern of *Ae. albopictus* was found to be different from the situation in Cameroon with *Aedes aegypti* predominating in urban areas, whereas *Ae. albopictus* is most prevalent in suburban and rural vegetated areas [58,59]. Nevertheless, a shift in *Ae. aegypti* distribution as a consequence of the invasion by *Ae. albopictus* has been reported in many places including Brazil, Florida [58,60,61], Puerto Rico [62] and the Mayotte island [63].

In the present study, out of 189 habitats found with larvae, *Ae. albopictus* was observed alone in 29.62% (56/189) and with *Ae. aegypti* in 61.37% (116/189) of the total habitats. These findings show the high frequency of the co-occurrence of the two species in nature. The ratio of *Ae. aegypti/Ae. albopictus* was 2.18:1 in rural settings, whereas this same ratio was approximately 1:25 in urban and peri-urban sites. These figures support the competitive superiority of *Ae. albopictus* to *Ae. aegypti* in urban and peri-urban sites. Similar observations in field and laboratory experiments have been made by previous studies, particularly in resource-limited conditions [49,64]. Yet the coexistence of *Ae. aegypti* and *Ae. albopictus* could be highly context-dependent and may depend on different factors including the specificity of aquatic resources or diet. A diet based on rapidly decaying resources, such as yeast, animal detritus or dead insects, was found to reduce competition between *Ae. albopictus* and *Ae. aegypti* and allowed their coexistence, while a diet based on deciduous or coniferous leaves was found to favour *Ae. albopictus* [49,64,65,66]. Seasonal variations alongside eggs desiccation was also found to affect the distribution and coexistence of both species [51]. Indeed, despite the fact that both species’ eggs could resist different environmental conditions, the drying of containers was found to be much more detrimental to *Ae. albopictus* eggs than to *Ae. aegypti* eggs in the Americas [51,67]. In studies conducted in Rio de Janeiro, Brazil, it appears that the end of the dry season was more favourable for *Ae. aegypti* than for *Ae. albopictus* immatures, which were found to be less abundant during this period [68]. It is not clear at this level whether a similar distribution pattern applies for both species in Cameroon.

Although at a global level, *Ae. albopictus* seems to be selecting for artificial containers when introduced to urban areas, it appeared that *Ae. albopictus’* preferred breeding habitats in rural sites were tree holes and leaf axils instead of artificial containers, which support a preference for natural sites in rural environments. Studies in Brazil and the western hemisphere also suggest *Ae. albopictus* has a high preference for natural containers in forested areas [68]. Competition between *Ae. aegypti* and *Ae. albopictus* could also negatively affect adult size, development rate, longevity and vectorial capacity [68]. Continuous surveillance should be carried out in Yaoundé to follow the evolution of these mosquito species, which are efficient vectors of arboviruses [25,26,27].

A low dispersion rate of *Aedes* species was recorded, with over 60% of the breeding places found between 0–10 m from human habitations. This could be attributed to the dependence of *Ae. albopictus* and *Ae. aegypti* upon humans as a source of blood meal. Indeed, *Aedes* species frequently colonize breeding and resting-habitats close to human dwellings. The high *Stegomyia* indexes observed in this study could result from the close proximity of both species to human dwellings as recently reported in the city of Yaoundé [55].

In regard to *Aedes* mosquito distribution in Yaoundé, the implementation of control strategies, such as the promotion of hygiene and the elimination of empty cans and containers around houses, has become important. The elimination of spare tires of vehicles or their collection and storage at specific sites in the city should also be envisaged to reduce *Aedes albopictus’* preferential breeding places. In rural settings, people should cover containers used for water storage to stop mosquito oviposition in these containers. Larval source management could also be implemented alongside campaigns to control *Aedes* populations.

## 5. Conclusions

*Aedes* mosquito species are largely distributed in different habitats of Yaoundé where they have colonized a great variety of water holding containers found around human dwellings. With the increasing number of arbovirus cases registered in Yaoundé, it is becoming urgent to implement control measures, such as larval source management, against these vectors to prevent the spread of arboviral diseases in Cameroon.

## Figures and Tables

**Figure 1 insects-12-00819-f001:**
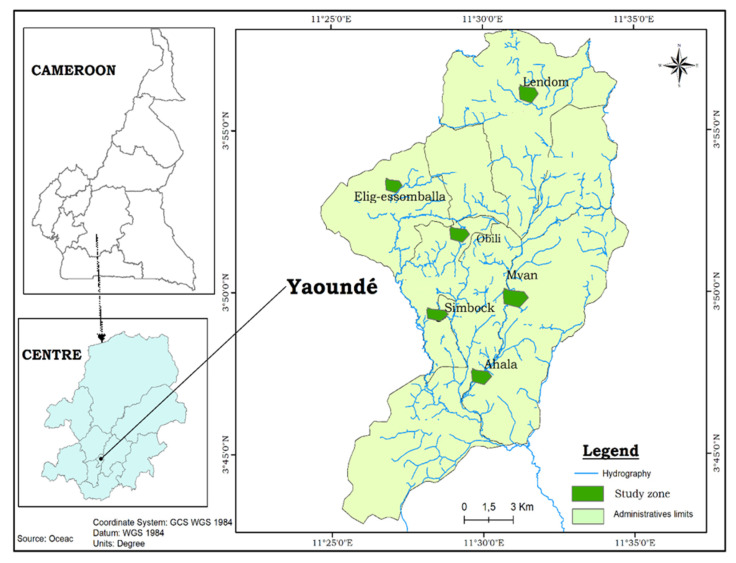
Map of the city of Yaoundé showing the study sites.

**Figure 2 insects-12-00819-f002:**
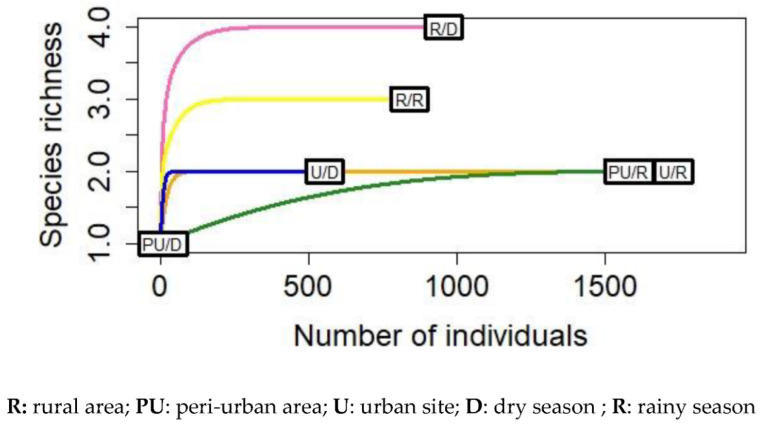
Rarefaction curves comparing *Aedes* mosquito abundance and species richness per season in each habitat type.

**Figure 3 insects-12-00819-f003:**
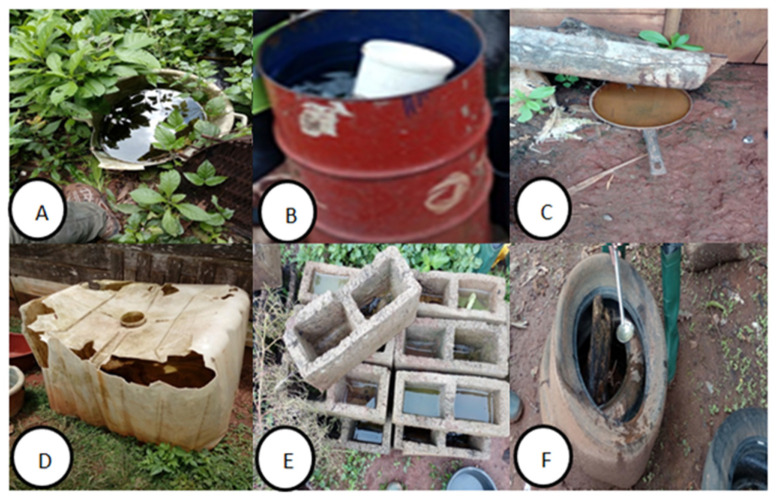
*Aedes* mosquito breeding habitats (**A**): plastic container; (**B**): tank; (**C**): metal; (**D**): plastic container; (**E**): cinder block; (**F**): tires.

**Figure 4 insects-12-00819-f004:**
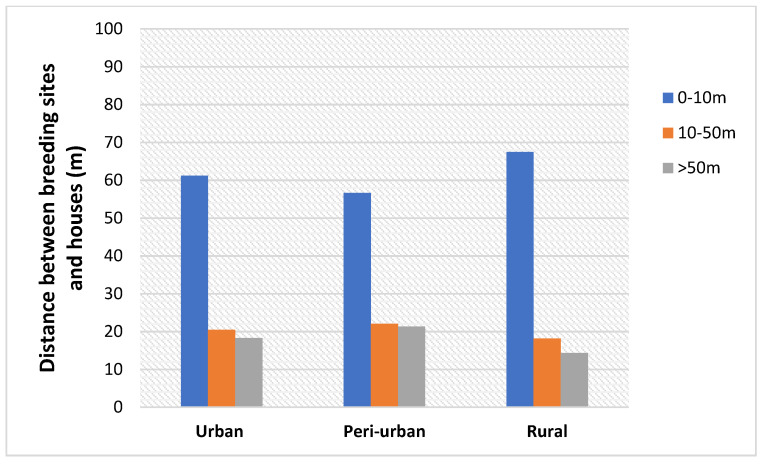
Distance between breeding habitats and houses.

**Table 1 insects-12-00819-t001:** Mosquito species identified across the transect from urban, peri-urban and rural areas.

Species	Ecological Zones						Total
	Urban		Peri-Urban		Rural		
	Obili	Mvan	Simbock	Ahala	Lendom	Elig-Essomballa	
*Ae. albopictus*	459	1755	1252	260	295	239	4260 (67.27%)
*Ae. aegypti*	3	68	71	7	588	577	1314 (20.75%)
*Ae. simpsoni*	0	0	0	0	65	20	85 (1.34)
*Cx. duttoni*	0	21	10	14	103	113	261 (4.12%)
*Cx. antennatus*	0	0	0	0	26	0	26 (0.41%)
*Cx. quinquefasciatus*	0	34	57	0	14	86	191 (3.01%)
*Lutzia tigripes*	0	2	1	0	8	9	20 (0.31%)
*Cx. (Culiciomyia)* group.	0	0	0	10	91	16	117 (1.84%)
*Anopheles funestus*	0	0	0	0	0	9	9 (0.14%)
*Eretmapodites* spp.	0	0	0	0	0	15	15 (0.23%)
*Toxorhynchites* sp.	0	0	0	0	1	0	1 (0.01%)
*Culex* spp.	0	0	12	0	2	6	20 (0.31%)
*Aedes* spp.	0	0	0	0	13	0	13 (0.20%)
**Total**	462	1880	1403	291	1206	1090	**6332**

**Table 2 insects-12-00819-t002:** Assessment of the correlation between sites breeding habitats, seasons and mosquito species abundance and distribution using generalized linear model.

	LR Chisq	Df	Pr (>Chisq)
Mosquito Species	22.18	3	*p* < 0.0001
Site (Urban/periurban/rural)	939.33	2	*p* < 0.0001
Season (dry/rainy)	108.13	1	*p* < 0.0001
Mosquito species vs. Site	484.4	6	*p* < 0.0001
Mosquito species vs. Season	0.81	3	0.8478
Site vs. Season	180.78	2	*p* < 0.0001
Mosquito species vs. Site vs. Season	198.02	6	*p* < 0.0001

**Table 3 insects-12-00819-t003:** Type of breeding habitats recorded with *Aedes* mosquito larvae in urban, peri-urban and rural settings.

Breeding Habitats		Urban		Peri-Urban		Rural		Total
		Obili	Mvan	Simbock	Ahala	Lendom	Elig-Essomballa	
**Tires**	**N(a)**	**20 (9)**	**33(7)**	**83(42)**	**19(5)**	**5(2)**	**8(4)**	**168(69**)
	*Ae. albopictus*	358	1350	1159	260	1	10	**3138**
	*Ae. aegypti*	3	68	62	7	15	47	**217**
**Plastic containers**	**N(a)**	**14(1)**	**7(2)**	**19(4)**	**2(0)**	**48(31)**	**47(30)**	**137(68)**
	*Ae. albopictus*	101	67	72	0	210	131	**581**
	*Ae. aegypti*	0	0	8	0	353	344	**705**
	*Ae. simpsoni*	0	0	0	0	65	2	**67**
**Metallic containers**	**N(a)**	**1(0)**	**1(0)**	**4(1)**	**0**	**10(9)**	**13(11)**	**29(21)**
	*Ae. albopictus*	0	0	21	0	30	27	**78**
	*Ae. aegypti*	0	0	1	0	109	94	**204**
	*Ae. simpsoni*	0	0	0	0	0	9	**9**
**Metallic tanks**	**N(a)**	**0**	**1(0)**	**0**	**0**	**6(6)**	**19(8)**	**26(14)**
	*Ae. albopictus*	0	0	0	0	10	34	**44**
	*Ae. aegypti*	0	0	0	0	65	13	**78**
	*Ae. simpsoni*	0	0	0	0	0	5	**5**
**Plastic tanks**	**N(a)**	**0**	**3(2)**	**0**	**2(0)**	**8(3)**	**4(4)**	**17(9)**
	*Ae. albopictus*	0	338	0	0	39	18	**395**
	*Ae. aegypti*	0	1	0	0	15	9	**25**
**Breeze block**	**N(a)**	**0**	**3(0)**	**0**	**2(0)**	**2(1)**	**3(2)**	**10(3)**
	*Ae. albopictus*	0	0	0	0	3	12	**15**
	*Ae. aegypti*	0	0	0	0	3	16	**5**
	*Ae. simpsoni*	0	0	0	0	0	4	**4**
**Tree hole**	**N(a)**	**0**	**1(0)**	**0**	**0**	**0**	**0**	**1(0)**
	*Ae. albopictus*	0	0	0	0	0	0	**0**
	*Ae. aegypti*	0	0	0	0	0	0	**0**
**Plant leaves**	**N(a)**	**0**	**0**	**0**	**0**	**0**	**1(1)**	**1(1)**
	*Ae. albopictus*	0	0	0	0	0	0	**0**
	*Ae. aegypti*	0	0	0	0	0	22	**22**
**Others**	**N(a)**	**0**	**3(0)**	**6(0)**	**0**	**5(4)**	**1(1)**	**15(5)**
	*Ae. albopictus*	0	0	0	0	2	7	**9**
	*Ae. aegypti*	0	0	0	0	28	32	**60**

N: total number of breeding sites inspected; (a): number of positives breeding sites; Others (sprayer, sink, wheelbarrow); plastic container refers to any container like plastic boxes, bowls, pans, without looking at their volume.

**Table 4 insects-12-00819-t004:** Co-occurrences of *Aedes* species and other species in the same breeding habitats in rural, peri-urban and urban sites in Yaoundé.

Species	Urban	Peri-Urban	Rural
*Ae. albopictus*	22.09% (19/86)	22.05% (30/136)	3.86% (7/181)
*Ae. aegypti*	0	0	14.91% (27/181)
*Ae. albopictus + Ae. aegypti*	3.48% (3/86)	10.29% (14/136)	12.15% (22/181)
*Ae. albopictus + Ae. simpsoni*	0	0	0.55% (1/181)
*Ae. albopictus + Ae. aegypti + Ae. simpsoni*	0	0	4.41% (8/181)
*Ae. albopictus + Ae. aegypti + Ae. simpsoni + Cx. culiciomayia group*	0	0	1.10% (2/181)
*Ae. albopictus + Ae. aegypti + Ae. simpsoni + Cx. Duttoni*	0	0	0.55% (1/181)
*Ae. albopictus + Ae. aegypti + Ae. simpsoni + Cx. Culiciomayia group + Cx. antennatus*	0	0	0.55% (1/181)
*Ae. albopictus + Cx. quinquefasciatus*	2.94% (4/136)	0.73% (1/136)	0.55% (1/181)
*Ae. albopictus + Cx. duttoni*	2.20% (3/136)	0.73% (1/136)	0.55% (1/181)
*Ae. albopictus + Lutzia tigripes*	0	0.73% (1/136)	0
*Ae. albopictus + Cx. Culiciomayia group*	0	0.73% (1/136)	0
*Ae. albopictus + Cx. Culiciomayia group + Cx. Duttoni*	0	0.73% (1/136)	0
*Ae. albopictus + Cx. Culiciomayia group + Cx. quinquefasciatus*	0	0	0.55% (1/181)
*Ae. albopictus + Cx. quinquefasciatus + Lutzia tigripes*	0.73% (1/136)	0	0
*Ae. aegypti + Cx. Culiciomayia group*	0	0	1.65% (3/181)
*Ae. aegypti + Cx. Culiciomayia group + Cx. quinquefasciatus + Lutzia tigripes*	0	0	0.55% (1/181)
*Ae. aegypti + Cx. duttoni + Lutzia tigripes*	0	0	1.10% (2/181)
*Ae.aegypti + Ae. simpsoni + Lutzia tigripes*	0	0	0.55% (1/181)
*Ae.aegypti + Cx. quinquefasciatus + Eretmapodites sp*	0	0	0.55% (1/181)
*Ae. albopictus + Ae. aegypti + Cx. quinquefasciatus*	0.73% (1/136)	(4/136)	1.65% (3/181)
*Ae. albopictus + Ae. aegypti + Cx. duttoni*	0	0.73% (1/136)	2.76% (5/181)
*Ae. albopictus + Ae. aegypti + Cx. antennatus*	0	0	0.55% (1/181)
*Ae. albopictus + Ae. aegypti + Lutzia tigripes*	0	0	0.55% (1/181)
*Ae. albopictus + Ae. aegypti + Cx. Culiciomayia group*	0	0.73% (1/136)	1.10% (2/181)
*Ae. albopictus + Ae. aegypti + Cx. duttoni + Cx. quinquefasciatus*	0	0.73% (1/136)	1.65% (3/181)
*Ae. albopictus + Ae. aegypti + Cx. Culiciomayia group + Cx. duttoni*	0	0	0.55% (1/181)
*Ae. albopictus + Ae. aegypti + Cx. Culiciomayia group + Cx. antennatus*	0	0	0.55% (1/181)
*Ae. albopictus + Ae. aegypti + Cx. duttoni + Lutzia tigripes*	0	0	0.55% (1/181)
*Ae. aegypti + Ae. albopictus + Eretmapodites*	0	0	1.10% (2/181)
*Ae. albopictus + Ae. aegypti + Cx. Culiciomayia group + Cx. quinquefasciatus + Lutzia tigripes + An. funestus*	0	0	0.55% (1/181)
*Ae. albopictus + Ae. aegypti + Ae. simpsoni + Cx. Culiciomayia group + Cx. quinquefasciatus*	0	0	0.55% (1/181)

**Table 5 insects-12-00819-t005:** Entomological *Stegomyia* indices in urban, peri-urban and rural settings.

Stegomyia Indices			
**Sites**	**Breteau Index (95% CI)**	**Container Index (95% CI)**	**House Index (95% CI)**
Urban	49.38% (41.07–57.69)	70.02% (67.64–72.41)	40.72% (40.38–41.07)
Peri-urban	40.27% (25–55.55)	42.71% (39.13–46.29)	41.94% (25–58.88)
Rural	42.31% (35.44–49.18)	34.62% (29.78–39.47)	44.91% (44.26–45.56)

95% CI: 95% confidence interval.

## Data Availability

All the data from the study is available in the manuscript.

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
