# Peer review of "Aedes Mosquito Distribution along a Transect from Rural to Urban Settings in Yaoundé, Cameroon"

_insects, 2021, doi:10.3390/insects12090819_

Round 1
Reviewer 1 Report
This is a good survey study of arboviral vectors around human habitation covering urban, peri-urban and rural settings. A few comments:
- Were Aedes aegypti mosquitoes examined for subspecies occurrence (A. aegyoti aegypti and A. aegypti formosus)? it would be interesting to see the subspecies distribution across habitat types and association with Aedes albopictus.
- In the rarefaction curves it is generally not valid to extrapolate the curves into hypothetical larger sample sizes. Especially with the relatively low overall species richness observed here, the extrapolations are just asymptotes and not really meaningful.
- Were Ae. aegypti and Ae. albopictus numbers affected equally by the difference between the rainy and dry seasons? Given their differences in larval container usage there may be differences there.
- In the discussion, considering the results of the manuscript, can the authors make some control strategy recommendations? This would be a good addition to the discussion.
Author Response
Please see the attachment." in the box if you only upload an attachment.

Reviewer 2 Report
I am attaching a Microsoft word file with my comments.

Author Response

(The authors gave the same response as above.)

Reviewer 3 Report
The manuscript determines the species, breeding sites and abundance of mosquito fauna in urban, peri-urban and rural sites in Cameroon. This study is of interest due the risk of arbovirus transmission in these areas.
I have a list of comments about the MS which I believe should be reviewed:
Title:
Why do the authors use the word "bionomics"? It includes mating, life-cycle, development...I suggest to remove this word since the study is focused in breeding sites and species abundance/richness.
Abstract:
Be careful with the italics. "spp." should be written in normal letter (not italics). Also check in the whole manuscript the genus names of Aedes or Culex since they should be written abbreviated after the first mention (e.g. lines 41, 185, 187, 236, 238..).
Introduction:
1. First sentence of the introduction is the same than in the abstract, please rephrase it.
2. Please check throughout the MS the names of the viruses: Zika, Chikungunya, Rift Valley, West Nile should be capitalized since are names of regions.
3. Lines 57-58 a reference is needed here
4. lines 58-68: I suggest to split the paragraph in two sentences
5. Lines 62-64: Is redundant with the previous paragraph. I suggest to join it with the previous lines
6. Lines 79-82: Here you are refering to the similar issue than in lines 57-65 but using other words.....
Materials and Methods:
1. Please include the UTM coordenates of each sites
2. Figure 1: What do Yaoundé 1, Youndé 2, Youndé 3.....Youndé 6 mean on the map?
3. How do you separate urban with peri-urban and rural? Did you analyse the % of houses and roads in X Km2 and then you used a threshold?
4. Why did you don't include natural sites?
5. More information is needed in the study desing: You performed a transect of how many meters? or you check the whole breeding sites of a fixed area of each site? Please explain it in "Study design"
6. Line 119: Which RNA later did you use? Please include manufacter and country.
7. Line 121: Aedes must be written in italics
8. Lines 126-127: I don't understand what does it mean. It seems that is considered a lower HI index for DENV transmission risk....please rephrase for better understanding.
Results:
1. Figure 2 needs better quality for publication. I also recommend to include the number of different Aedes species. No mosquitoes were collected during dry season in peri-urban areas?
2. Figure 3 should be improved, picrures C and F seems to be narrower than the others...
3. Lines 190-191 co-ocurrence between different genera is also more frequent in rural areas.
4. Figure 4: Should be improved using the same font and size letter than the MS and removing the corner lines.
5. I cannot finde Toxorhynchites in table 3, please, check it.
6. Stegomyia of table 4 should be written in italics and check the word "indices"
7. Where are the results of the GLM between habitats?? tables 3 and 4 have only shown descriptive analysis with numbers and percentages but no results of GLM.
Discussion:
This part should be strongly improved and a deep discussion of results is needed.
- Seems that you are describing again the results (e.g.lines 217-225 and so on).
- Line 226: where is "elsewere"? In Africa or also in other continents?
- In general you need to compare with results from other continents
- Which are te consequences of the displacement of Ae. aegypti by Ae. albopictus? Increase of different virus transmission? Please discuss it and include examples in other countries/continents
- Line 237: The introduction of Ae. albopicus in late 90's? You wrote 2001 in the introduction. Please check it.
- Be careful with the term "competition". Did you check if thy compete for food? please don't confuse competition with co-existence in the same breeding site
- Lines 255-257: Why do you think that these genera breed in artificial containers due to the absence of preferencial ones? Maybe the same for Ae. aegypti and Ae. albopictus ? How can you prove it?
- I miss some discussion regarding the vector role of the other species collected (e.g Cx. quinquefasciatus, An. funestus) and the role of Toxorhynchites as biological control in breeding sites
References: Please check the italics of species names
Author Response
"Please see the attachment." in the box if you only upload an attachment.

Round 2
Reviewer 3 Report
The revisions have been properly addressed so I accept the reviewed MS